# The Use of the Internet of Things in the Distribution Logistics of Consumables

**Karol Kvak and Martin Straka \***

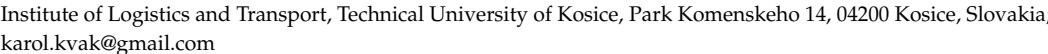

Institute of Logistics and Transport, Technical University of Kosice, Park Komenskeho 14, 04200 Kosice, Slovakia; karol.kvak@gmail.com
* **Correspondence: martin.straka@tuke.sk**

**Abstract:** This research focuses on integrating Internet of Things (IoT) technologies in the field of distribution logistics of consumables, specifically on the optimization of dispensing through vending machines in a corporate environment. The primary goal was to identify IoT-based vending machine solutions that increase the work efficiency and cost-effectiveness of consumables distribution. In the first part, this paper focuses on the definition of IoT, and its current trends and development over time. With the help of the authors, we define the key aspects of its use in practice, and we also focus on establishing critical security points. We evaluate its impact on the industry, where there is currently a great pressure for automation and robotization of production, as a result of which the labor market is experiencing a major transformation of job positions from executive to technologically creative ones. Using the performance measurement method, this study established key performance indicators (KPIs), which we then collected and, through thorough research, determined the basis for setting the future state. Using the method of technological innovation, we prepared prototypes of the vending machine, which were then placed at the customer's location for a 3-month trial period using the method of technological demonstration. During the trial period, key performance indicators were collected again using the performance measurement method, and we then evaluated them using the comparison method. The results demonstrate a significant reduction in the cost of consumables and a simultaneous increase in employee efficiency. These results underline the potential of IoT technologies in the field of distribution logistics for consumables. The research points to the advantages and perspectives of using the Internet of Things in the business environment, especially in the field of distribution logistics.

**Keywords:** Internet of Things; distribution logistics; vending machines; system; efficiency

## 1. Introduction

The aim of this research was to identify vending machine solutions for consumables based on Internet of Things (IoT) technologies and to create a functional model that is usable for businesses. The motivation of this research is the ongoing pressure on companies to increase the efficiency of employees' work and reduce the costs of consumables. The research was carried out with the help of a specific expenditure of consumables in the company.

The development of the human population has been accompanied by technological development from the beginning. In the past, these were mostly accidental discoveries that pushed humanity forward and helped it survive. With increasing technological possibilities and the desire of man to progress, the intensity of development also increased. New technologies are becoming part of our everyday life, and we mostly do not even have time to perceive everything that science will bring us. Gradually, we began to meet terms such as computer, Internet, mobile phone, server, and digitalization.

The digitization process is a large-scale project whose goal is to create a new information network of society, controlled by information and communication technologies that

find, collect, process, and disseminate information through global telecommunication networks. We are currently observing the so-called "Fourth Industrial Revolution" (Industry 4.0). The term was introduced in 2011 as part of the Hi-Tech strategy in Germany. Industry 4.0 is characterized by the development of data exchange and automation, and includes the introduction of cyber-physical systems, the Internet of Things, the cloud, computing technology, the use of unmanned vehicles, and 3D printing [1]. Technologies have been gradually introduced into the everyday life of a person, such as housing, school, sports, eating, and entertainment. However, we perceive the most extensive benefit in the work area, where we use technology to increase the efficiency and quality of work [2]. With regard to the digital economy through the prism of production, it can be noted that the introduction of new materials, the transition to new technologies for the production of goods, the automation of production processes, and the application of innovations in the field of logistics are the main development trends [3]. The need for their further development is therefore essential for further growth. Based on the topic that we address in this publication, we discuss the Internet of Things (IoT) in more detail. It is a part of almost every technological innovation.

### 1.1. Key Views of the Authors on the Definition of IoT

The term IoT and its concept were introduced in 1999 by Kevin Ashton—the founder of the Auto-ID research group at the Massachusetts Institute of Technology. The latter defines it as an integrator of surrounding objects into a single network that exchange information with each other in real time, without human intervention [4]. Objects that are integrated include technologies such as wireless communication (Wi-Fi, Bluetooth, etc.), microelectronic systems (intelligent sensors and chips), and, of course, the Internet. With their help, a huge amount of data is transferred all over the world. The author of Tran-Dang, Hoa [5] writes in his publication that the vision of the Internet of Things (IoT) allows multiple embedded devices, objects, and people with limited resources to connect through the Internet Protocol for ubiquitous data exchange. Logistics is seen as a key player aiming to achieve full visibility and transparency by leveraging ubiquitous interconnectivity to collect reliable and secure real-time data. In addition, valuable information extracted and transformed from IoT data can be used to create intelligent services and applications to improve logistics activities as well as the overall performance of logistics operations [5]. Figure 1 shows the dramatic development and authors' interest in IoT, where we show the number of publications available on the topic of IoT in the years 2000–2023.

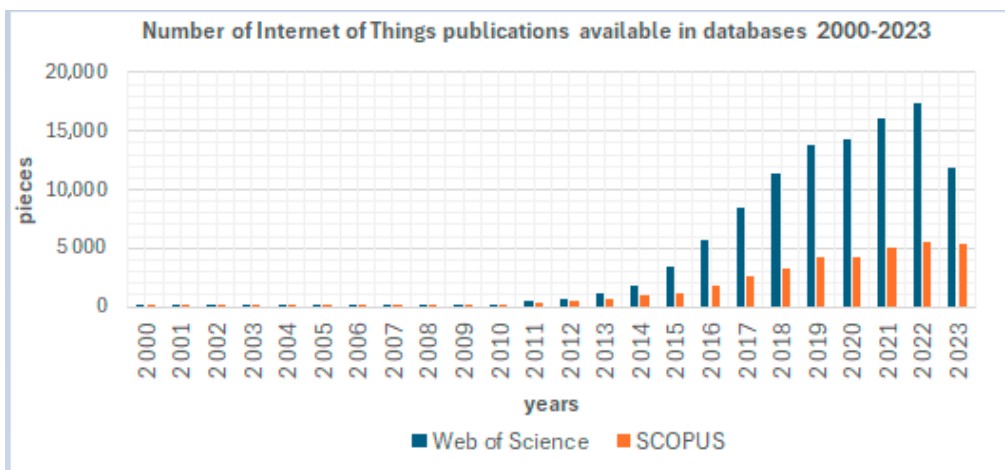

**Figure 1.** Number of Internet of Things publications available in databases 2000–2023.

Authors describing advanced IoT technology differ in their definitions only by the point of view of the industry that influences the author. In their publication, Ali Zainab et al. and their team define it as the ability of things and devices around us to connect with each

other with elements of intelligent behavior and management [6]. More recent publications state that it is about creating an ecosystem for mutual communication and collaboration between physical devices and the Internet to improve efficiency, automate processes, and provide new opportunities for various industries, such as healthcare, industry, logistics, and manufacturing [7]. The author uses the term ecosystem as a metaphor for the expression of the word base or basic element. The main idea that technology companies are playing with is the creation of a worldwide infrastructure of interconnected physical things that would provide connectivity anytime and anywhere [8]. Gradually, more and more cyber components are integrated into physical systems and thus include a larger spectrum of things around us. The basic pillars on which IoT is built are:

- Sensors and devices—Their task is to collect various data from their surroundings. These are, for example, thermometers, motion sensors, GPS devices, or sensors for measuring humidity, light, and sound, up to vital function meters [9].
- Connection to the Internet—Devices must be connected to the Internet, which allows data to be transferred to and from these devices [10]. The connection can be implemented through various technologies, including WiFi, Bluetooth, 3G, 4G, and 5G, but also special protocols for IoT (for example, LoRaWAN and NB-IoT.)
- Cloud solutions—Data from these devices are usually sent to cloud servers, where they are stored, processed, and analyzed. These solutions enable centralized access to data and their further use [11].
- Data processing and analysis—An important part is the ability to process and analyze a large amount of data generated from these devices [12]. It may include machine learning techniques and data analysis to extract useful information.
- Feedback and control—Based on data analysis and intelligent algorithms, IoT devices can respond to changes in the environment and perform various tasks, including controlling other devices or generating alerts and messages.
- Applications and user interfaces—Data and analysis results can be presented to users through applications or other user interfaces [13]. This allows users to monitor and control connected devices and processes [14].

Figure 2 shows how Rafiul Kabir graphically illustrates a virtual prototype for modern IoT applications in his study. The center of the entire system is technology (software), which processes and cooperates with individual IoT elements such as various sensors, processes, and systems. They communicate through communication networks, and individual data points are recorded in data repositories. The data are then analyzed and processed using modern elements of artificial intelligence. In most cases, these technologies are managed by companies in various fields of industry.

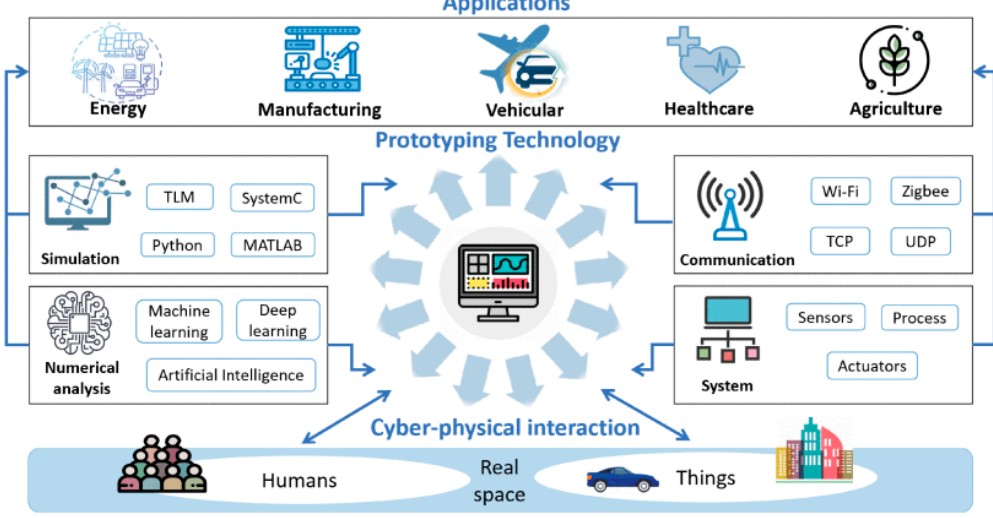

**Figure 2.** Virtual prototyping for modern IoT applications [7].

*1.2. IoT Security Challenges*

However, many authors also point to the pitfalls of using IoT and ask whether we are mentally prepared for such extensive use of IoT in practice and whether the legislation in the field of its use is sufficiently set [15]. The authors Singh, S. and Singh, N state in their publication that one of the aspects is the physical security of devices, since physical access to devices can allow unauthorized persons to manipulate them [16]. This mainly concerns the threats of theft, sabotage, or physical damage to the equipment itself. Currently, these threats are solved by placing devices in inaccessible places or by monitoring these devices using camera systems or a guard service. Another important aspect of security is communication protection, which includes secure communication between individual IoT devices and cloud servers [17]. These threats include eavesdropping, man-in-the-middle attacks, and data falsification, which compromise the confidentiality and integrity of transmitted data [18]. Other authors describe the solution of managing identities and access rights as an important task, using various technologies and procedures. These include, for example, identification and authentication, securing communication protocols, defining access rights, and their regular revocation. We ensure that only authorized persons have access to important data sources [19]. The devices are part of the secure networks through which they communicate. Here the question is whether their integration with these networks using firewalls is sufficient and secure [20]. In many cases, IoT devices are being misused to create botnets (networks of compromised devices) for mass cyberattacks. There are many types of cyberattacks that can target different aspects of information security and cybersecurity. The most common cyberattacks include malware (infecting a device with a harmful virus), phishing (obtaining sensitive information such as passwords, and personal or bank details), DoS and DDoS (overwhelming a network or target system, leading to unavailability of services), Man-in-the-Middle attacks (modification of communication between two parties by a third party), SQL injection, spoofing, and many others.

There is therefore an imperative to protect against the misuse of devices for these cyberattacks through standardization and compliance, which help create uniform practices and minimize security risks [21]. Organizations should monitor and adapt to current security standards and regulations. Therefore, they create risk assessment methodologies. Figure 3 shows an example of the risk assessment methodology.

The human factor can be the weak link in IoT. Users and administrators should be familiar with IoT security practices and certifications to minimize human risks. In further development, the creation of standards and interoperability in IoT is inevitable, as they enable effective integration and cooperation within the entire ecosystem. Industry, standardization organizations, and government institutions should work together to create open, secure, and interoperable standards [22].

With the gradual development of information technologies, normative frameworks and standards have also evolved, helping organizations and entities create and implement effective strategies for information security and data protection in accordance with international practices and legal requirements. Compliance with these standards is important to ensure the credibility, integrity, and confidentiality of information, and to reduce risks associated with cyber threats and data breaches. Some of the most well-known normative frameworks and standards include:

- ISO/IEC 27001: This standard defines requirements for information security management in organizations. It includes a set of controls and procedures that organizations can implement to ensure an adequate level of information security.
- ISO/IEC 27002: This document provides a detailed set of recommendations for managing information security. It contains specific control measures and procedures that organizations can implement to protect their information.
- NIST Cybersecurity Framework: Developed by the National Institute of Standards and Technology (NIST) in the USA, this framework provides a set of best practices for managing cybersecurity. It helps organizations identify, protect, detect, respond to, and recover from cyber threats.

- GDPR (General Data Protection Regulation): This European regulation sets rules for the protection of personal data of EU citizens. It establishes requirements for data protection, informed consent, transparency, and the rights of individuals regarding their personal data.
- HIPAA (Health Insurance Portability and Accountability Act): This US law relates to the protection of personal health information and sets requirements for the security and confidentiality of health information.
- PCI DSS (Payment Card Industry Data Security Standard): This standard defines requirements for the security of payment cards and payment systems to minimize the risk of data theft involving payment cards.

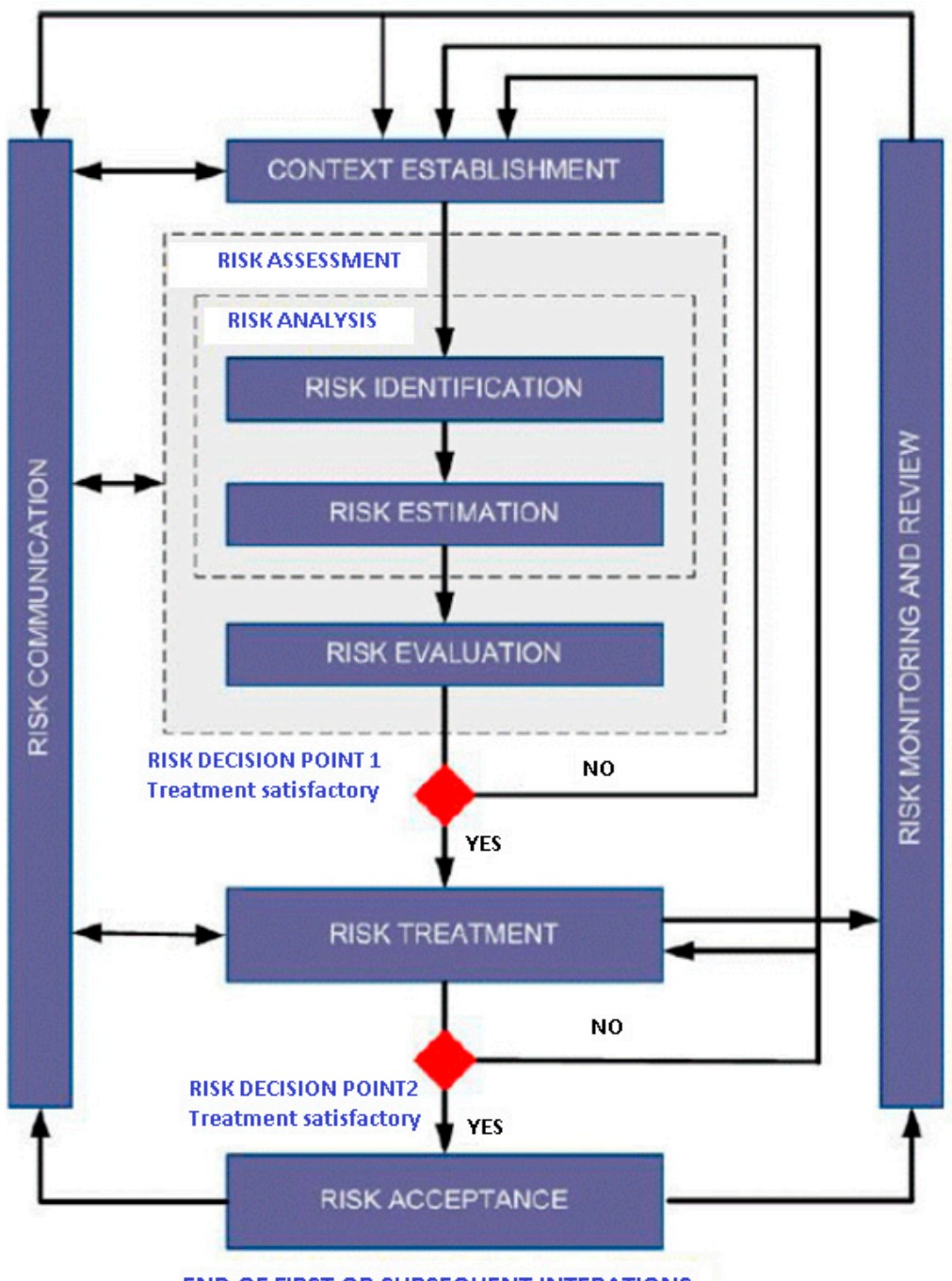

**Figure 3.** Risk assessment methodology in IT security.

*1.3. The Impact of IoT on Industry and the Labor Market*

IoT has a significant impact in the industrial sector, affecting aspects of production, operations management, and the overall functioning of industrial enterprises. It helps industry in the automation and optimization of its processes, using the possibility of real-time monitoring and control, data analysis, and support in the decision-making process [23]. Automated lines minimize the need for employees and thus reduce costs. Predictive maintenance, on the other hand, enables monitoring and collection of data on the state of equipment, on the basis of which we can predict the need for maintenance and thus prevent production downtime. With the help of energy sensors, we can effectively manage energy consumption in real time, which once again helps businesses to minimize costs and achieve energy neutrality. All the above-mentioned aspects are aimed at increasing effectiveness or cost reductions [24]. There are also aspects that have an impact on increasing the quality of processes (products) and reducing the requirement for financial capital in the form of inventory management [25].

An example of a successful implementation of IoT in industry is General Electric (GE), which uses sensors and IoT technologies to monitor the health of its production equipment and systems, such as turbines, motors, or other complex industrial systems. These sensors collect data on temperature, pressure, vibrations, energy consumption, and other parameters of operation. These data are then analyzed using advanced analytics tools and machine learning algorithms to predict potential equipment failures or outages before they actually occur. Such an approach allows GE to perform maintenance based on condition and need, rather than a traditional planned maintenance schedule. This minimizes unplanned outages, extends the life of equipment, reduces maintenance costs, and increases the overall efficiency of operation.

As we mentioned above, automation significantly affects the change in the labor market. As a rule, companies try to automate tasks that are regularly repeated or are time-consuming and reduce the need for manual workers. At the same time, work efficiency increases, which means that we need a significantly lower number of hours worked for the same work. Several authors build on this basis, evaluating IoT and other technological changes very negatively [26]. They describe negatives such as the loss of jobs due to the replacement of manual work with automated systems, increasing social differences and the concentration of income in the hands of large technological companies, building dependence on technology, reducing morale among employees due to fears of job loss, and many others [27]. Of course, we must perceive all aspects as justified and built on a real basis. However, there are authors who invite technological skeptics to debate and point out that with the growing number of technological devices, there are demands for experts in the development, implementation, and maintenance of IoT systems. A huge quantity of data is produced, which needs to be analyzed, revealing meaningful information and trends [28]. This creates new segments of the labor market and a number of new jobs oriented towards professional work. However, the transformation of work skills from manual to digital is a necessity. Therefore, teaching in schools is changing, where the development of digital and analytical skills, and professional training for working with these technologies, comes to the fore [29]. The development of the educational system in the field of IoT in education has undergone considerable development and progress. These changes include:

- Introduction of new curricular contents: With the advent of IoT in industry and everyday life, school curricula have begun to adapt. Curricular content has been expanded to include topics related to sensors, connected devices, data analytics, and cybersecurity to prepare students for the needs of modern industry.
- Introduction of new learning materials and tools: Software and hardware manufacturers have developed special educational tools and platforms aimed at teaching IoT. These tools allow students to experiment with IoT devices, create and test their own projects, and gain practical experience with IoT technologies.
- Expanding curricula and career opportunities: Schools have begun offering expanded IoT-focused educational programs that allow students to gain a deeper understanding

of the topic and gain hands-on experience. These programs may include courses, workshops, competitions, and internships in industrial enterprises.

- Expansion of certifications and accreditations: With the increase in demand for IoT professionals, certification programs and accreditations have begun to expand. Students can earn certificates and diplomas in the field of IoT, which helps them get a job in industry or continue their university studies.

The same process must be implemented in the transformation of existing employees, where companies must invest funds in their training. Increasing the efficiency of production saves financial resources, which can be invested in the quality of the working environment, increasing the expertise of employees and the standard of living. In addition to the readiness of employees for the arrival of IoT technologies, companies must thoroughly analyze the existing infrastructure and technological means that they will subsequently use for the introduction of IoT technologies. This includes, for example, the expansion and improvement of computer networks, the increase in the technical level of computer systems, and the software openness of existing systems. These systems must be thoroughly secured (the method of securing is described in Section 1.3). Finally, the company must have the prerequisite of openness of the company's management towards new investments in the field of development of the use of IoT technology, changes in the company's strategy, and the existence of a strategy for the implementation itself, as it is a complex and long-term development process. The created strategy is subsequently adapted to the communication with internal employees and the investigation of the existence of partnerships and cooperation with external suppliers and consultants who can help with the implementation of IoT, and thus provide professional help and support.

IoT can contribute to solving social and environmental problems in the fields of public health, ecological sustainability, or management of urban systems [30].

An important outcome of the discussions must be the quality setting of regulations and standards that will manage the risk associated with IoT technologies. This will ensure the protection of privacy and security of the company. At the same time, the gradual introduction of technologies is necessary, which will give society room for understanding and adaptation [31].

### 1.4. The Impact of IoT on Distribution Logistics

Distribution logistics is an area of logistics that deals with managing the flow of goods from their place of production or storage to the place of consumption or sale. This process involves planning, organizing, managing, and monitoring the various activities associated with the distribution of goods to ensure their efficient and reliable transfer and delivery from one point to another. Distribution logistics includes activities such as warehousing, packaging, transportation, inventory handling, order management, inventory tracking, and delivery of goods to customers. The goal of distribution logistics is to optimize these processes in order to ensure fast and efficient delivery of goods, minimize costs, and maximize customer satisfaction. In the current business environment, where customer expectations are often high and competition is strong, distribution logistics plays a key role in the success of businesses. Effective management of distribution processes enables companies to achieve a competitive advantage through faster deliveries, lower costs, and better customer satisfaction.

The development of distribution logistics has undergone major challenges in recent years. Consumers have high expectations of their suppliers. They require fast deliveries after sending the order, accurate information about its status, flexibility of delivery options, a simple and intuitive way of purchasing, up-to-date information about the stock status, flexibility and the possibility of exchanging or returning goods, and, finally, they expect that the offers they will receive they will be personalized to their requirements [32,33]. These requirements are often a key factor that suppliers must pay attention to in order to maintain customer loyalty and create a positive shopping experience [34,35].

IoT is having a significant impact in distribution logistics, transforming traditional practices and bringing new opportunities to improve efficiency, tracking, and management. It helps businesses meet customer expectations without the negative impact of these investments. With the help of the most modern technologies, in distribution logistics, we can monitor the current state of stocks, and predict the need for purchase in connection with the season and the expected behavior of the consumer [36,37]. GPS modules help us optimize routes, and loading and unloading in real time, thereby increasing the efficiency of carriers. We can use them to track the location of the vehicle, traffic conditions, and other relevant factors, which can reduce the delivery time [38]. At the same time, the consumer can check and monitor the state of equipment and delivery of their shipment. We can predict the estimated delivery time almost exactly to the minute [39,40].

Another important group of devices that use IoT technologies are robots and robotic systems that help us increase the efficiency and speed of distribution logistics and minimize the risk of errors. These are, for example, autonomous mobile robots that are able to independently navigate in warehouses and perform a whole range of tasks, such as transporting supplies within the warehouse and storing them, collecting goods in the warehouse for specific orders, completing and packing orders, and loading pallets on the distribution pickup. Robots can also help with the maintenance and cleaning of warehouses. These devices use a number of IoT technologies, such as ultrasonic sensors and infrared sensors used to detect obstacles around the robot and its navigation, pressure sensors used to measure pressure when handling objects, temperature sensors used to ensure the correct storage temperature, and sensors of volume, lighting, etc.

In the future, a very promising form of improvement in distribution logistics is the use of drones, which use a large number of IoT technologies. They should especially help us in the final phase of distribution, by delivering the shipment itself. This is supposed to be a very effective way of distribution, especially in hard-to-reach and isolated areas such as mountain huts, islands, or remote villages. This is a new trend of efficient, fast, accurate, and ecological delivery.

It can therefore be concluded that the Internet of Things is a key factor for effective distribution logistics. Its ability to provide accurate inventory tracking, optimize routing and transport management, quickly recognize and resolve problems, ensure delivery quality, and provide transparent data for analysis improves the performance of distribution processes [41–43]. The benefits of IoT in distribution logistics not only increase efficiency, but also service quality and customer satisfaction, thereby contributing to the overall success and competitiveness of businesses in today's rapidly changing business environment. In the next part of the study, we explore and suggest how IoT technologies can improve the management and tracking of the distribution of consumables, increase efficiency, reduce costs, and improve the overall logistics efficiency of the enterprise.

## 2. Materials and Methods

The aim of this study was the implementation of new knowledge and discoveries of the use of IoT technologies in machines for dispensing consumables in manufacturing companies, which will make the dispensing of consumables more flexible and efficient, reduce the costs of material consumption for companies, and make the material more available to employees. At the same time, the distribution system of deliveries to the company is optimized and automated, thereby reducing distribution costs and the need for cash flow for the supplier of consumables. During the study, it was found that the issue had not been investigated by another author in the past, and therefore we consider its implementation to be beneficial.

The research was carried out on a case study in the production company SkyBlue, which employs more than 1300 employees and has been operating in the Slovak market for more than 20 years. The company produces sheet metal bumpers for cars, which requires a high consumption of consumables such as protective gloves, glasses, cutting tools, and marking material. The company's total turnover is EUR 48 million and it produces more

than 18,000 bumpers a year. The company was selected based on open communication by the company and an effort to participate in research by making data available.

The research was carried out in cooperation with the company DistriLog, which supplies consumables to industrial enterprises in Slovakia. It employs 80 employees and its annual turnover is EUR 16 million. The company supplies businesses with various logistics solutions, where there is a high potential for optimizing distribution logistics using IoT solutions. Distribution logistics solutions are the company's main competitive advantage, and their development can further support its stable position as a market leader.

In a specific case, we use the performance measurement method to analyze the current state of issuing consumables. Using the performance measurement method allows us to focus on specific research goals, using the selection of specific KPIs (key performance indicators) and metrics that are relevant for evaluating the effectiveness of IoT implementation in vending machines. In our case, the key performance indicators include the cost of expended consumables. The results of the measurement are thus objective, minimizing the subjective view of the research. We were able to collect and use the obtained data during the entire course of the research. They thus helped us in making decisions and evaluating the real impact on the project. This part also included the use of the method of time studies, with the help of which we measured the time required by an employee to issue consumables.

Based on the results of the current state and the customer's requirements, we used the method of technological innovation and the subsequent method of technological demonstration to create a prototype vending machine using IoT technologies, which was implemented at the customer's location. We chose the technological demonstration method despite its financial demands. However, this method gave us a realistic view of how IoT technology might work in a real distribution environment. This is important when simulating practical conditions. The demonstration allowed us to verify the working efficiency of the IoT implementation, and we could watch over time how it affects processes and immediately reveals potential challenges and benefits. Interaction with stakeholders helped us solve problems on the spot.

The above-mentioned performance measurement methods and the technology demonstration method gave us the basis for using the comparison method. We can compare selected KPIs before and after the introduction of IoT technologies, which provided us with measurable indicators of the impact of the technology. Comparing data allowed us to identify changes and trends, quantify the impact, and identify the causes and consequences of changes, which have us room to understand the connections that contribute to improvement or, conversely, to deterioration.

We supported the research results using the method of opinion research, where we evaluated the opinions of interested parties. These enabled us to obtain qualitative information about the attitudes, experiences, and evaluations of people who are directly involved in or affected by the implementation of IoT. This can provide a deeper insight into the subjective perception of technology. Opinion research complements the quantitative data obtained from performance measurement methods and provides context and explanations for why certain quantitative patterns may exist. Subsequently, we gained a comprehensive understanding of user needs and preferences with the use of IoT. This can be valuable in customizing the implementation according to real needs. Finally, it helped us to support the participation and involvement of stakeholders in the implementation of the process.

In Figure 4, we graphically illustrate the methodology for introducing a new solution for issuing consumables:

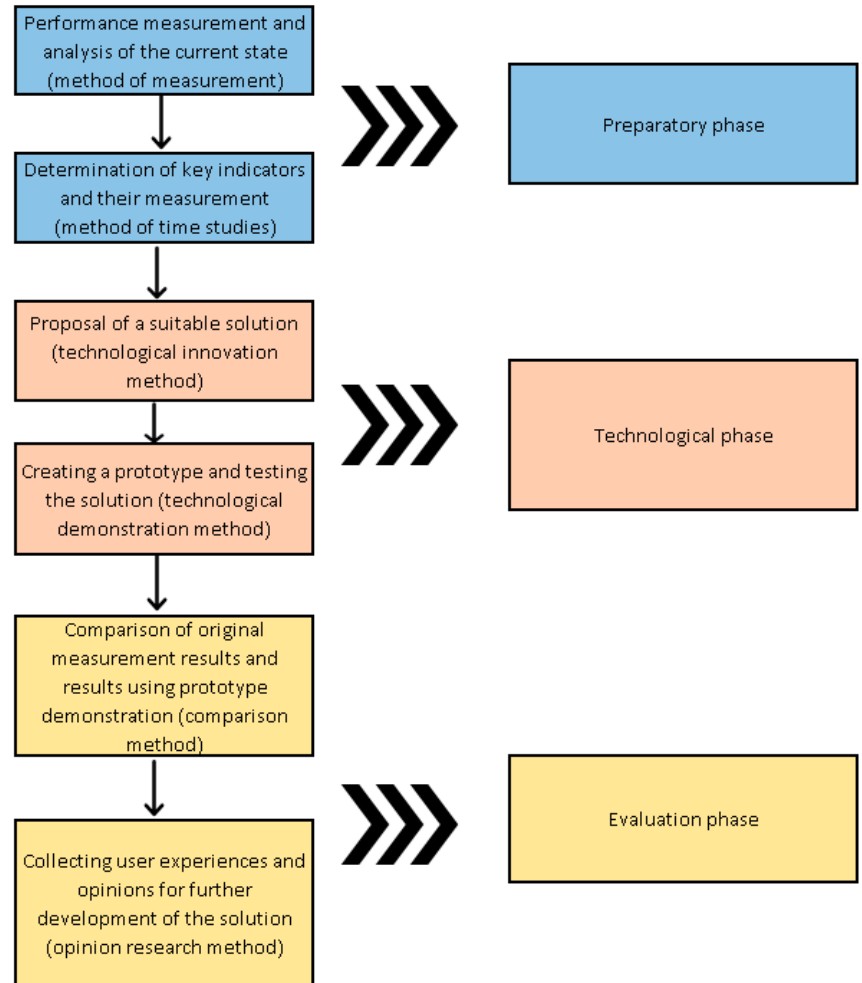

**Figure 4.** The methodology for introducing a new solution for issuing consumables.

At the end of this paper, we propose another procedure that needs to be implemented for further research in the field of using IoT technologies in distribution logistics and specifically in the issue of consumables in companies.

## 3. Results

By describing the research methodology for the implementation of new knowledge and discoveries of the use of IoT technologies in machines for dispensing consumables in manufacturing companies, we obtained a procedure for the implementation of research in the SkyBlue company. Individual steps were implemented with the participation of DistriLog employees.

### 3.1. Consumable Dispensing System before the Introduction of IoT

Before the introduction of the vending machine with IoT technologies, the system of issuing consumables took place in the SkyBlue company using a time-consuming method of issuing from the central warehouse. Based on the data provided (consumables used), SkyBlue registered a total of 193 items consumed in 2022 with a total value of EUR 296k. However, we found that only 7% of the items made up 76% of the total value of the material consumed. In this phase of the research, we focus on the items with the highest turnover for the year 2022. The remaining 93% of the items can be the subject of research in the next scientific work. The mentioned 7% of items can be seen in Table 1.

**Table 1.** Items consumed at SkyBlue.

| | Price per Item (EUR) | 1–12/2022 | Total Value 2022 (EUR) | 1–6/2023 | Average/Month | Monthly Consumption in EUR |
|---|---|---|---|---|---|---|
| Cutting gloves A4587 size: 6 | 4.19 | 2526 | 10,584 | 1312 | 213 | 893 |
| Cutting gloves A4587 size: 8 | 4.19 | 10,874 | 45,562 | 5314 | 899 | 3768 |
| Cutting gloves A4587 size: 10 | 4.19 | 12,311 | 51,583 | 6580 | 1050 | 4397 |
| ESD gloves C 1283 size: 6 | 2.32 | 5820 | 13,502 | 2714 | 474 | 1100 |
| ESD gloves C 1283 size: 8 | 2.32 | 8412 | 19,516 | 5478 | 772 | 1790 |
| ESD gloves C 1283 size: 10 | 2.32 | 315 | 731 | 156 | 26 | 61 |
| Safety Glasses R41 | 3.17 | 3814 | 12,090 | 1796 | 312 | 988 |
| Respirator 9322+ | 2.98 | 12,965 | 38,636 | 7200 | 1120 | 3338 |
| Transparent tape 50 mm | 7.32 | 2478 | 18,139 | 1547 | 224 | 1637 |
| High visibility tape 25 mm (Yellow) | 9.45 | 1518 | 14,345 | 941 | 137 | 1291 |
| Marker pen (black) | 2.13 | 312 | 665 | 205 | 29 | 61 |
| Marker pen (red) | 2.13 | 38 | 81 | 24 | 3 | 7 |
| Marker pen (white) | 2.13 | 15 | 32 | 9 | 1 | 3 |
| Total | | | 225,466 | | | 19,335 |

In Table 1, in addition to the name of the items, we can also see the prices for the items, the amount consumed for the year 2022, and the total value that the company paid for the individual items. For a higher reporting value of the data, we also requested the consumption of items for the months January to June in 2023. Subsequently, we calculated the monthly average and the average monthly value that the company invests in consumables in the company.

The monthly consumed amount and the value of the consumed amount served as basic KPIs for the use of other research methods [35].

Using the observation method, we created a process map that precedes or defines the issue of consumables. The individual steps of the process are shown in Figure 5. The process begins with a request from the employee, who writes out a paper request for the allocation of consumables, which must be registered in the ERP (enterprise resource planning) system after approval by the superior. According to the availability in the warehouse, the employee receives confirmation with the information that he can pick up the item in the central warehouse, which is located 142 m from the production hall. If the item is not available at the central warehouse, the ordering process is started with the supplier. The unavailability of products at the central warehouse is fundamental, as it can lead to a complete stoppage of production or prevents the employee from entering the workplace, as the employer was unable to provide him with personal protective equipment (hereinafter referred to as PPE). Availability is therefore seen by the company as key. Despite this, the company recorded 74 cases where the employee did not receive the required consumables and therefore could not be allowed to work.

An important KPI in this process is the determination of the time taken by an employee to issue consumables. The calculation of this KPI was carried out using the method of time studies on a sample of 200 employees. Table 2 shows the results, and describes the maximum, minimum, and average times. From the initial analysis, we found extreme values; therefore, we additionally supplemented the calculation of the median, which was used in further calculations and analyzes.

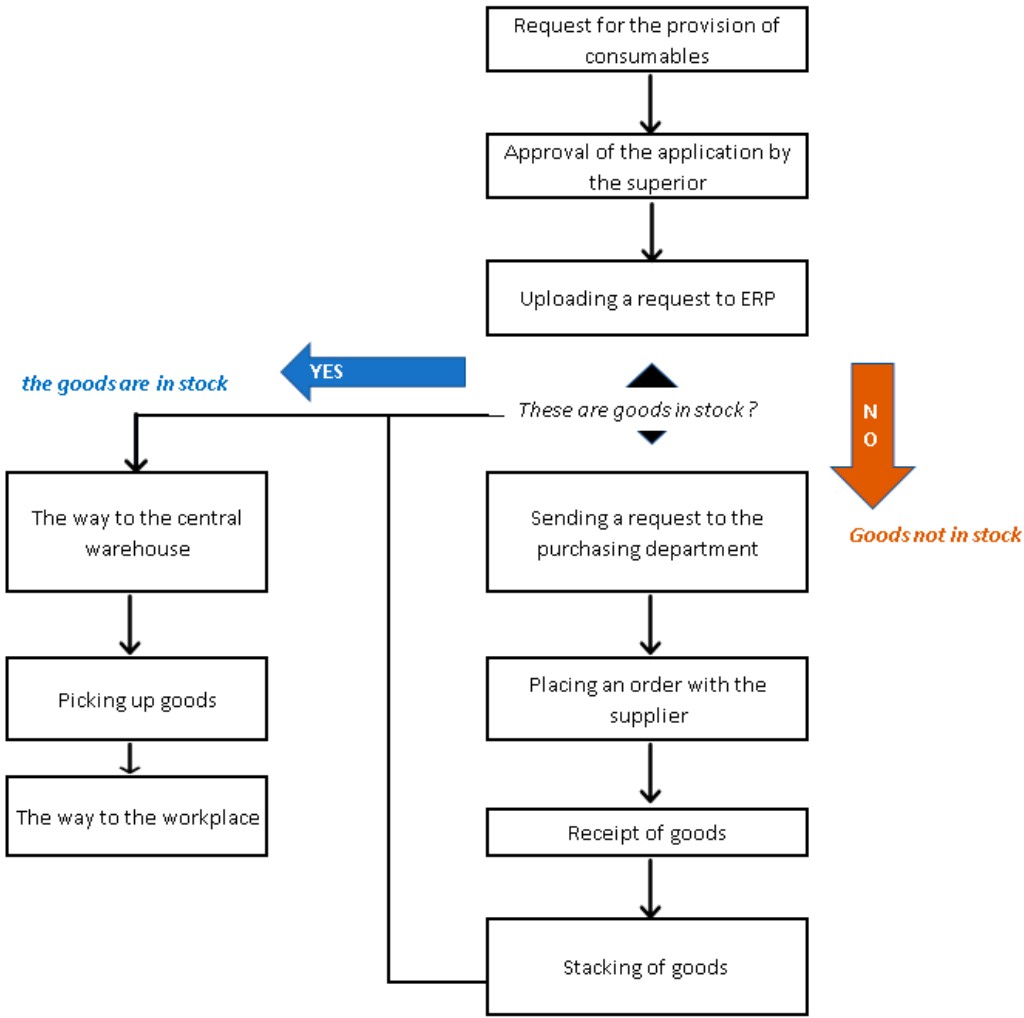

**Figure 5.** Consumable dispensing process (internal resource).

**Table 2.** Time to dispense consumables without IoT.

|  | Maximum (Minutes) | Minimum (Minutes) | Average (Minutes) | Median (Minutes) |
|---|---|---|---|---|
| Request for the provision of consumables | 7 | 2.12 | 3.54 | 2.68 |
| Approval of the application by the superior | 2 | 1.04 | 1.54 | 1.12 |
| Uploading a request to ERP | 3 | 0.95 | 1.42 | 1.29 |
| The way to the central warehouse | 14 | 2.54 | 5.47 | 3.45 |
| Picking up goods | 4 | 0.45 | 1.12 | 0.95 |
| The way to the workplace | 12 | 2.47 | 5.89 | 3.84 |
| Total | 42 | 9.57 | 18.98 | 13.33 |

The results of the previous research were discussed with the management of SkyBlue and, based on the discussion, the basic requirements that must be taken into account when designing the solution were defined:

- Products must be available to employees at the workplace, so that there are no transfers of employees.
- Products must be available to employees 24/7.
- The disbursement system must be simple for employees.

- It is necessary to minimize the need for other employees (storekeepers, buyers) in the entire process of issuing.
- Records of issued items must be personalized (preferably up to the employee).
- Expenditure must be subject to claims so that costs are not exceeded.

### 3.2. Design of a Machine for Dispensing Consumables with IoT Technology

The second phase of the research was to use the method of technological innovation to find a suitable vending machine system using IoT technology that would be suitable for dispensing consumables in the company. We carried out this research in cooperation with the company DistriLog, which in the past dispensed consumables using string vending machines, which are used to dispense sweets or drinks (Figure 6).

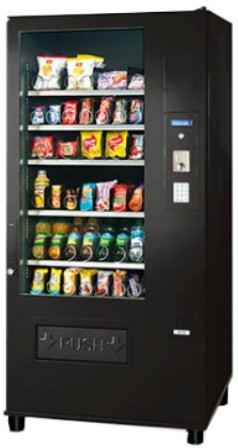
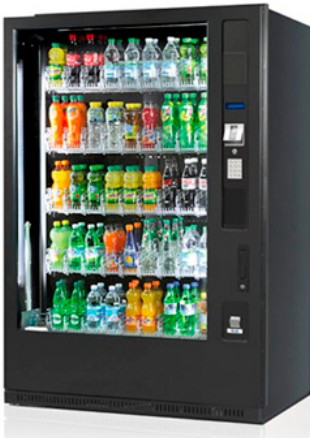

**Figure 6.** Vending machines for drinks and sweets.

However, the currently used solution covered only part of the SkyBlue customer's requirements, and failed to meet the requirement of personalized records of the issue of consumables and control of the consumption of the issued material. At the same time, the solution was demanding for the support staff, as the customers refilled the vending machines themselves, which was very time consuming (the material needs to be specially packaged for the correct functionality of the machine).

Together, we visited a vending machine manufacturer with a request for the production of an "intelligent" vending machine using IoT technology. The basis for fulfilling the requirement was the insertion of a computer unit that was used to process and store information in the vending machine, and communicate with other vending machines and with the administrative system. Since vending machines are usually located in production areas, they must be resistant to external influences (humidity, temperature, dust, and dirt). The most suitable solution was the choice of an industrial computer, as shown in Figure 7.

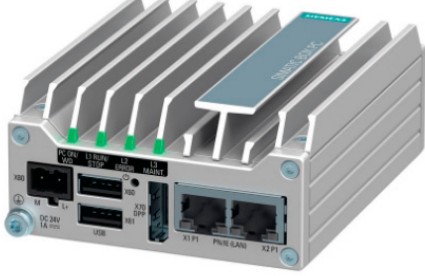

**Figure 7.** Computer installed in vending machines.

Another important task that had to be solved was the communication of this device with the external environment, as companies do not allow the addition of third-party devices to their network for security reasons. It was therefore necessary to use an external modem (Figure 8), which was placed near the vending machine, and solved the online communication with the external environment.

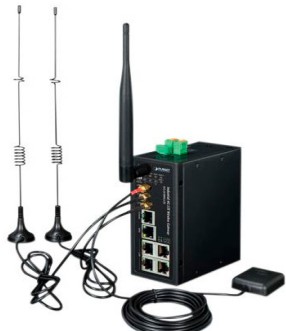

**Figure 8.** 4G industrial modem for external communication.

Among the other IoT technologies that needed to be implemented in the vending machine system was an RFID chip reader (Figure 9), which ensured the personalization of consumables per employee. Each employee was assigned a unique RFID chip, which was used to identify themselves upon arrival, departure from the workplace, or in the company's catering facilities [40]. All data are stored on a cloud server for data security, which is regularly backed up and protected by the latest protections. The vending machine supplier's programmers programmed the vending machine management software, and also the user software that sets the parameters of the vending machine and the dispensing from it.

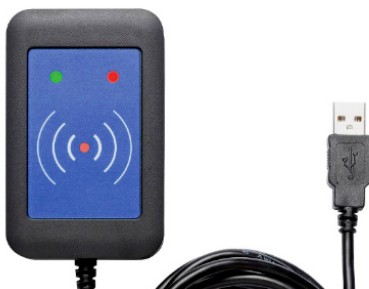

**Figure 9.** RFID reader used in vending machines.

The result of the innovation process was a vending machine (Figure 10) with IoT technologies ready for demonstration testing at the customer's location.

The vending machine was designed to collect data on the consumption of individual employees and manage the vending machine. The collected data form an important element in the further activities of the supplier of consumables, on the basis of which he is able to propose customized solutions to the customer. Table 3 shows an example of output data from the vending machine. The employee ID is recorded here, which is linked to the employee's name. The name of the employee cannot be displayed due to the Personal Data Protection Act. Together with the employee ID, the model of the consumables taken, the date of the issue, and the time of issue are recorded. An employee can choose only one piece of product per login; therefore, each selection is registered as a separate transaction. The material consumption report can be downloaded from the machine only by the system administrator based on personalized login data. After logging in, the administrator chooses the time period for which he wants to download the data. The resulting file is in xlsx format.

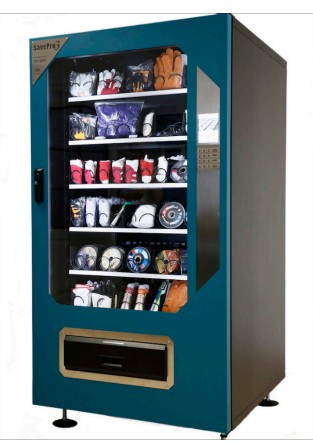

**Figure 10.** Vending machine with IoT technology.

**Table 3.** Output of data stored in vending machines.

| Employee ID: | Item Delivered: | Date: | Time: |
|---|---|---|---|
| 001 | Cutting gloves A4587 size: 6 | 3 March 2024 | 8:02:00 |
| 002 | ESD gloves C 1283 size: 8 | 3 March 2024 | 8:32:00 |
| 001 | Respirator 9322+ | 3 March 2024 | 8:37:00 |
| 004 | Cutting gloves A4587 size: 6 | 3 March 2024 | 8:38:00 |
| 065 | Cutting gloves A4587 size: 6 | 3 March 2024 | 8:45:00 |
| 006 | Marker pen (red) | 3 March 2024 | 9:15:00 |
| 034 | ESD gloves C 1283 size: 10 | 3 March 2024 | 9:21:00 |
| 008 | ESD gloves C 1283 size: 10 | 3 March 2024 | 9:32:00 |
| 112 | ESD gloves C 1283 size: 10 | 3 March 2024 | 9:37:00 |
| 010 | Cutting gloves A4587 size: 6 | 3 March 2024 | 9:38:00 |
| 012 | ESD gloves C 1283 size: 8 | 3 March 2024 | 10:17:00 |

A separate important part was securing the system and data against various forms of cyberattacks, which we described in the previous parts of the study. Together with the development engineers, a risk methodology for cybersecurity for vending machines was developed. Unfortunately, as it is an internal document related to security, this methodology could not be published in the study. However, it is based on the methodology shown in Figure 3. Security against various forms of attacks mainly consists of the security firewall used, which monitors and controls the network data flow to and from the vending machine. The firewall thus prevents unauthorized access and protects it from various forms of attack. Another form of security used is antivirus software, which was also installed in the industrial computer of the vending machine. This software helps to detect and subsequently remove various malware software threats that could compromise system security. The operating system and individual software parts of the vending machine are regularly updated, which contributes to the removal of known security errors and vulnerabilities. The data itself are encrypted on the disk to prevent their misuse. Two-factor authentication of the administrator is required for access to the data, using a password and subsequent mobile authentication. The last element of security is the decentralized backup of data using a random key so that it is impossible to predict this backup. With the help of the backup, the possibility of data recovery in the event of an attack is ensured. The security proposal against physical attack is ensured by the proposed camera system and the installation of the vending machine on the floor of the building.

### 3.3. Implementation Process and Operation Process

The third phase of the research on the dispensing of consumables using vending machines with IoT technologies was to use the demonstration method to place vending machines in the SkyBlue company, on which we monitored the development of the monitored KPIs. The company DistriLog, in cooperation with the manufacturer of vending machines, installed five vending machines at the customer SkyBlue. The number of vending machines was designed based on the monthly consumption of individual items and the capacity of the vending machines. Vending machines were installed in frequented places within the production hall to be accessible to employees.

The installation process consisted of several parts. The basic part was the creation of a database of employees in the vending machine, where each employee was assigned a unique code under which they were subsequently kept in the vending machine system. The RFID chip number was assigned to the unique code, with the help of which the employees could subsequently select consumables. This unique code also ensured compliance with the GDPR, to which employees are entitled.

Employees were divided into groups according to cost centers (job classification), on the basis of which it was possible to collectively determine the requirements for consumables and set limits for individual positions, which prevented them from over selection. With this step, we eliminated the need for approval of each selection by superiors. The database of employees and products can be changed by the user as part of normal traffic. However, only the person authorized to perform this change has this authority.

In the next part of the installation process, the delivery process was defined and all employees of the company were also retrained. The issuing process is shown in Figure 11. This has been significantly simplified and consists only of moving to the vending machine and selecting the consumables itself.

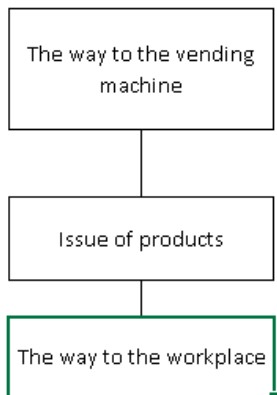

**Figure 11.** Consumable dispensing process with IoT technologies.

Refilling of vending machines and taking care of vending machines is under the direction of DistriLog. Vending machines and the goods in them are the property of this company until the moment they are issued to employees. SkyBlue pays a monthly rent for the administration and rental of the vending machine. The result of the activity is one invoice for the goods consumed in a given month and one invoice for the rental of vending machines.

Research using the demonstration method ended after 3 months of operation, during which the set KPIs were analyzed. The resulting KPIs are shown in Table 4, which shows the times of the individual phases of the expenditure. As we can see, the process was shortened from the original 13.33 min to 2.4 min. By making consumables available directly at the workplace, the influence of external factors that influenced employees' choice of consumables was reduced. Employee performance increased, which had an immediate impact on meeting company-wide production standards.

**Table 4.** Time to dispense consumables with IoT.

|  | Maximum (Minutes) | Minimum (Minutes) | Average (Minutes) | Median (Minutes) |
|---|---|---|---|---|
| The way to the vending machine | 6 | 0.23 | 1.12 | 0.86 |
| Issue of products | 1 | 0.45 | 0.65 | 0.51 |
| The way to the workplace | 7 | 0.24 | 1.24 | 1.03 |
| Total | 14 | 0.92 | 3.01 | 2.4 |

Table 5 shows the average consumption of monitored items. Using the comparison method, we found that consumption was reduced by 15.5%. The reduction in consumption can be justified by personalizing individual expenses and setting limits for individual positions. In the event that the SkyBlue company did not have an established approval of the expenditure of consumables in the past, we estimate possible savings at the level of 20–25%.

**Table 5.** Comparison of consumption before and after the introduction of IoT.

|  | Price per Item (EUR) | Average/Month before IoT | Monthly Consumption in EUR before IoT | Average/Month after IoT | Monthly Consumption in EUR after IoT | Monthly Cost Saving in % |
|---|---|---|---|---|---|---|
| Cutting gloves A4587 size: 6 | 4.19 | 213 | 893 | 169 | 708 | 20.7% |
| Cutting gloves A4587 size: 8 | 4.19 | 899 | 3768 | 772 | 3235 | 14.2% |
| Cutting gloves A4587 size: 10 | 4.19 | 1050 | 4397 | 883 | 3700 | 15.9% |
| ESD gloves C 1283 size: 6 | 2.32 | 474 | 1100 | 425 | 986 | 10.4% |
| ESD gloves C 1283 size: 8 | 2.32 | 772 | 1790 | 614 | 1424 | 20.4% |
| ESD gloves C 1283 size: 10 | 2.32 | 26 | 61 | 20 | 46 | 23.6% |
| Safety Glasses R41 | 3.17 | 312 | 988 | 266 | 843 | 14.7% |
| Respirator 9322+ | 2.98 | 1120 | 3338 | 987 | 2941 | 11.9% |
| Transparent tape 50 mm | 7.32 | 224 | 1637 | 183 | 1340 | 18.2% |
| High visibility tape 25 mm (Yellow) | 9.45 | 137 | 1291 | 112 | 1058 | 18.0% |
| Marker pen (black) | 2.13 | 29 | 61 | 21 | 45 | 26.9% |
| Marker pen (red) | 2.13 | 3 | 7 | 3 | 6 | 12.9% |
| Marker pen (white) | 2.13 | 1 | 3 | 1 | 2 | 25.0% |
| Total |  |  | 19,335 |  | 16,335 | 15.5% |

We subsequently added the identified KPIs to the created profit analysis (Table 6), which gave us an overall view of the introduction of vending machines in the company.

**Table 6.** Profit analysis.

| DistriLog Profit Analysis | | | | |
|---|---|---|---|---|
| **Customer:** | **SkyBlue** | | | |
| Reduction in consumption | Yearly | | Yearly | 3 years |
| Annual consumption of the material you want to place in the vending machine | EUR 225,000.00 | | | |
| Minimum reduction in percentage | 15.5% | | | |
| Quantity of vending machines | 5 | pcs | | |
| Total annual savings | EUR 34,875.00 | | EUR 34,875.00 | EUR 104,625.00 |
| Soft costs (time reduction for workers) | | | | |
| Time saved per employee in operation | 8.52 | min/day | | |
| The number of employees who take material from the vending machine (average per day) | 514 | | | |
| Average hourly wage of an employee + benefits | EUR 10.38 | | | |
| Total annual employee savings | EUR 181,827.71 | | EUR 181,828 | EUR 545,483 |
| Soft costs (time reduction for purchaser and stockkeeper) | | | | |
| Time for vending machine management | 1 | hour | | |
| Time saved by buyer per week | 8 | hour | | |
| Buyer's hourly wage plus benefits | EUR 12.84 | | | |
| Storekeeper's time saved per week | 57.8 | hour | | |
| Storekeeper´s hourly wage plus benefits | EUR 9.21 | | | |
| Total annual savings for the buyer and storekeeper | EUR 25,654.94 | | EUR 25,655.00 | EUR 76,965.00 |
| Total savings | | | EUR 242,358.00 | EUR 727,073.00 |
| Vending machines rental fee | | | | |
| Monthly rent for one vending machine | EUR 480.00 | | EUR −28,800.00 | EUR −86,400.00 |
| | | | EUR 213,558.00 | EUR 640,673.00 |
| Annual savings | | | EUR 213,558.00 | EUR 640,673.00 |

In the profit analysis (Table 6), we can see that, just by introducing vending machines, our direct costs per year decreased by almost EUR 35,000. Even more interesting, however, is the result of increasing the efficiency of the consumption of consumables and the subsequent saving of costs through the saved time of employees during the issue, which was worth EUR 181,827 per year, and cost savings worth EUR 25,654 associated with saving the time of buyers and storekeepers, by which we completely eliminated the work of ordering and expenses. Even after taking into account the costs associated with the rental of vending machines and their management, the annual cost savings for the SkyBlue company came to EUR 213,558. The company therefore decided to launch this project into full operation, with the temporary interest of expanding the portfolio to other types of goods.

In the last phase of the research, we used the method of collecting opinions to ask SkyBlue employees how satisfied they are with the introduction of the service of issuing consumables through a vending machine with IoT technologies.

*"I am very satisfied with the current release. Consumables are available at any time and close to my workplace. I have more time for my work."* Employee No. 1 (production operator)

*"The choice is easy, I don't have to fill out any papers and go beg someone."* Employee No. 2 (production operator)

*"I have a machine 5 steps from my workplace. I always have the necessary material available and I don't have to wait for the storekeeper."* Employee No. 3 (production operator)

*"We are very satisfied with the supply of consumables. I always have employees at my workplace and they don't wander around my warehouse. Protective equipment is always available to them, so nothing prevents them from using it."* Employee No. 4 (foreman)

*"Finally, a system that suits us. With vending machines, we save more than €200,000 a year, we meet the production plan and the employees are at their workplaces. At the same time, I can see consumption per specific workplace and employee, which allows us to better plan production costs."* Employee No. 5 (production manager)

*"Since the introduction of vending machines, I have no worries about purchasing consumables. I take care of 75% of the consumables, where I issue one summary order per month. I can even import it into our ERP based on the report from the vending machine. We are already working with the supplier DistriLog on other solutions that would help us automate the material distribution logistics."* Employee No. 6 (purchaser)

*"The project to introduce vending machines was conducted very professionally from the beginning. The installation process took place without any problems with emphasis on the customer's needs. Vending machines help us to fulfill the economic result and set KPI (cost reduction, production fulfillment plan, increasing efficiency). The return on investment is immediate."* Employee No. 7 (Supply chain director)

*"We will gladly support any project that helps us fulfill our vision. We are open to new challenges, and vending machines were one of them. Until now, I have not encountered anything similar."* Employee No. 8 (CEO)

## 4. Discussion

Many authors describe the benefits and use of IoT technologies in everyday life. As we discussed in the introductory part of the study, the perception is different. However, research in practice shows us that we are far from fulfilling the potential of use, which can be decisive for the further perception and direction of these technologies. However, we can say with certainty that it is a technological, social, and economic mistake. We can see the same perception of IoT in the field of logistics, and specifically distribution logistics. We demonstrated the possibility of using IoT technology in the automation of this process on a simple, everyday process of issuing consumables in the company. The new process setting thus met the customer's requirements, which were to increase the availability of consumables to employees, ensure the demand for the material, increase the efficiency of the expenditure, and reduce the costs of the expenditure of consumables. As we can see from the final profit analysis, even in this case of using IoT technology, we managed to significantly reduce costs and simplify the process [33]. We can conclude that we have fulfilled the goal of the research. With the help of the implementation of new knowledge and discoveries of the use of IoT technologies in machines for dispensing consumables in manufacturing companies, the dispensing of consumables has become flexible and efficient and the cost of material consumption for the company has been reduced, so it is more accessible to employees.

At the same time, the research highlighted to us the need for further investigation of the possibilities of using IoT technologies in distribution logistics, and specifically in vending machines for consumables. There is a need for further investigation of the impact of the introduction of these technologies for the supplier of consumables and its impact on the process of refilling the vending machine.

In further research, researchers should focus on the financial impact of these changes for suppliers of vending machines, as increased costs for replenishing consumables are expected. At the same time, the question arises as to how to issue consumables that are not suitable to be placed in the vending machine due to size limits. During the research, we encountered several limits, which included the size limit of vending machines. We solved them by specifying suitable consumables that can be placed in the machine. Key

limitations include the openness of the company's management and employees in bringing new technologies into their business, and the effort and opportunity to invest. We therefore recommend future researchers should carefully choose the company where they will carry out their research [40]. The pure limitation for us was the selection of a supplier for the production of vending machines, where it was necessary to combine their knowledge of hardware and the ability to find a software solution. We managed to solve this element by finding external suppliers of programming services.

## 5. Conclusions

The implementation of the Internet of Things (IoT) in the distribution logistics processes of consumables, specifically through vending machines, has proven to be an effective way to reduce costs and increase productivity. The results showed a significant improvement in the processes of issuing consumables, which testifies to the potential of IoT technologies in the field of business operations [40].

The methodological approach, which included methods of performance measurement, demonstration, comparison, and opinion research, enabled a comprehensive evaluation of IoT implementation. This knowledge can be valuable for companies looking for ways to make their distribution processes more efficient and optimize the use of consumables. In conclusion, IoT technologies have a significant impact on distribution logistics, and their integration can lead to significant benefits, including cost reduction and efficiency gains. These conclusions suggest that IoT investments in distribution logistics could be justified for businesses looking for innovative solutions and a competitive advantage. However, a big challenge is the development of security measures to preserve cybersecurity, as the introduction of IoT technologies creates space for misuse of data and cyberattacks that can threaten the rights of people, businesses, and society.

**Author Contributions:** Conceptualization, K.K. and M.S.; methodology, K.K. and M.S.; software, K.K.; validation, K.K. and M.S.; formal analysis, K.K. and M.S.; investigation, K.K.; resources, K.K.; data curation, K.K; writing—original draft preparation, K.K.; writing—review and editing, M.S.; visualization, K.K.; supervision, M.S.; project administration, M.S.; funding acquisition, M.S. All authors have read and agreed to the published version of the manuscript.

**Funding:** This research was funded by the grant number project KEGA 010TUKE-4/2023 Application of educational robots in the process of teaching the study program industrial logistics.

**Data Availability Statement:** The original contributions presented in the study are included in the article, further inquiries can be directed to the corresponding author.

**Conflicts of Interest:** The authors declare no conflicts of interest. The funders had no role in the design of the study; in the collection, analyses, or interpretation of data; in the writing of the manuscript, or in the decision to publish the results.

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
