# Peer review of "The Use of the Internet of Things in the Distribution Logistics of Consumables"

_applsci, doi:10.3390/app14083263_

Round 1

Reviewer 1 Report

Comments and Suggestions for Authors

The aim, purpose and object of the research is missing.

Image 1 missing source. Did the authors come to these conclusions on their own, which software did they use to analyze the publications, which bases did they include, based on which they came to this knowledge? Why is the period from 2007 to 2015 covered, and not the period until 2023? The image has insufficient resolution, it is necessary to insert a new image. Each chart should be accompanied by a table showing how many publications were actually published in the observed period.

Line 78/79/80 is not quoted correctly. Instead of the word team, it is necessary to put et al.

Line 82 mentions the word ecosystem. It is necessary to define what is meant by the term ecosystem.

It is necessary to clarify Figure 2. What exactly does it refer to, what does it represent? What is the connection between the image and the theme of the work?

Chapter 1.2. it is necessary to refine and add norms from the domain of information security that will complement the topic of IoT security. Test protocols, penetration tests should be mentioned. Line 125, it is necessary to define exactly which authors state the above, not to speak in general terms. The sentence in line 127 is unclear, it needs to be reworked. Line 133/134 needs to be defined exactly which standards are involved. It is unclear whether these are ISO standards or others. Chapter 1.2. it should be completed with risk assessment, i.e. risk analysis and risk assessment methodology.

Chapter 1.3. it is necessary to define how IoT affects the industry (give an example). In the chapter, it is necessary to review the development of the education system. In addition, the readiness of the organization to implement technologies such as IoT was not considered, i.e. it does not talk about the technical and technological readiness of implementation, but only about people. It is necessary to broaden the perspective in order to obtain a more systematic overview.

In chapter 1.4. it is necessary to describe more clearly how IoT affects distribution logistics since the general impact of GPS, i.e. the Internet on connectivity and the ability to track deliveries in real time is described. Based on the description, it is not possible to draw a conclusion about the impact of IoT on distribution logistics, as stated in line 204. In this chapter, it is necessary to describe the way of mutual communication between different devices via IoT, to mention robotization, drones, robots that can be used for delivery and similar to.

Chapter 2. It is necessary to graphically present the methodology. It is not clear what the research aims to achieve, have any of the other researchers conducted this or similar research? KPIs are mentioned but not clearly articulated.

Table 1 must be a table, not a picture. Line 304 mentions ERP but did not define what the abbreviation means. Image 3 must be of better quality. It is not a uniform font. Image source is missing. Table 2 needs context, how was the specified time collected, in what period was it collected, what was it measured by?

The term KPI is overused. I think its meaning is misunderstood. Throughout the research, it would be good to clearly define what it is about.

It is necessary to define the term distribution logistics and describe what it entails.

There is a lack of guidance for future researchers. It is necessary to describe the limitations of the research and how the limitations of the research were overcome.

Comments on the Quality of English Language

It is recommended to read the text once more and correct potential mistakes.

Reviewer 2 Report

Comments and Suggestions for Authors

The study aims at incorporating Internet of Things (IoT) technologies into the distribution logistics of consumable goods, particularly enhancing the efficiency of vending machines in corporate settings. The main objective is to discover IoT-enabled vending machine strategies that boost the distribution efficiency and cost savings for consumables.

The following deficiencies were observed, it is suggested to improve the text of the paper as follows:

  - in Figure 1, old data is used. It is definitely necessary to replace the image with updated data that includes, in addition to the data for 2024, and the projection of the future number of users

  - in chapter 1.2. The authors of IoT Security Challenges open up the topic of security challenges well, but I am of the opinion that what has been stated needs to be supplemented, considering the significant detected weaknesses of IoT devices. In this context, misuse and taking control of IoT devices in creating DDoS attacks must be mentioned. Additionally, the possibilities of applying blockchain technologies in improving the security of IoT applications is certainly an area that should be mentioned.

I am free to cite examples of state-of-the-art works from that area (of course without the author's obligation to cite said works, but only when showing the direction of improving the text)

10.1109/JIOT.2021.3090909

doi.org/10.3390/electronics13040687

10.1109/ACCESS.2024.3361310

  - please do not use the color red in data display, in all tables

- please improve picture 3 - the arrows are missing and a clear understanding of the process that is to be shown

- think about whether picture 8 is necessary. I also suggest that pictures 5, 6, 7 and others be shown differently - as equipment integrated into the device shown in picture 8. All of this can be shown in one picture (diagram).

- picture 9 seems unnecessary from reviewer point of view

- user satisfaction research does not need to be presented in a long way. It is not appropriate to give specific answers about subjective experience.

- in the conclusion chapter, it is not appropriate to refer to the literature [40]

- in conclusion, address the security challenges of applying the IoT concept in this domain

- additionally, add a few sentences about future work in this area.

Reviewer 3 Report

Comments and Suggestions for Authors

The reviewed paper deals with the application of the IoT paradigm in the field of consumer goods distribution. The paper presents several sections and paragraphs. In addition, the paper presents an application case study to support the proposed framework. 

The work is very interesting, and it denotes a commitment on the part of the authors to develop the application part. Nevertheless, there are a few unclear things that should be improved and developed in order to make the work more usable:

- It is advisable to emphasise more within the introductory part the objectives of the work presented.

- Figure number 1 refers to a period that does not reflect current research, it is advisable to update the reference to more recent research, some of which I recommend:

10.1007/978-3-031-31066-9_65

10.1016/j.jmsy.2021.03.005

10.1007/s12652-020-02521-x

- Reword the introductory sentence of the chapter 'Materials and Methods', I cannot start with the results of the study if I have not yet been presented with the study, perhaps the authors meant the objectives?

- In order to improve the chapter "Materials and Methods", it would be appropriate to propose a figure describing the proposed architecture in detail. 

- In sub-chapter 3.2 Design of a machine for dispensing consumables with IoT technology, an important part should be devoted to the data acquisition system, the sensors used, and the type of data used, and how this data is used to justify the data. 

Comments on the Quality of English Language

The text is fluent and needs no special comment.

Round 2

Reviewer 1 Report

Comments and Suggestions for Authors

I suggest accepting the work as it is.

Author Response

The reviewer does not require any modifications.

Reviewer 3 Report

Comments and Suggestions for Authors

I would like to thank the authors for the improved version of the work presented, and to a large extent the suggestions made have been fully taken on board by the authors.  For this reason, for the purpose of a final revision of the work, it is recommended to further improve chapter 3.2, as it is not yet fully clear what type of data they take into consideration, e.g. the datasheet and the type of sensors used could be described.

In addition, figure no. 5 from the new numbering should be figure no. 6.

Another area to be improved is the security part of the system, the authors have described the possible scenarios very well in the introductory part, but in the part they have developed, it is not well emphasised how this issue is dealt with by the authors.

For this reason, it is recommended to implement the following parts before the final version.

I would like to thank the authors for their work and wish them a successful continuation of their work.

Author Response

Response to Reviewer 3 Comments

Point 1: It is recommended to further improve chapter 3.2, as it is not yet fully clear what type of data they take into consideration, e.g. the datasheet and the type of sensors used could be described. In addition, figure no. 5 from the new numbering should be figure no. 6.

Response 1: Chapter 3.2 has been supplemented with information on the structure of recorded data, including the possibility of an output report. we also added a table with an example of recorded data. We rechecked all figures and adjusted their numbering.

Point 2: The security part of the system, the authors have described the possible scenarios very well in the introductory part, but in the part they have developed, it is not well emphasised how this issue is dealt with by the authors.

Response 2: A section was added to chapter 3.2, which deals with securing the vending machine against possible forms of attacks
